# Molecular insight into interactions between the Taf14, Yng1 and Sas3 subunits of the NuA3 complex

Minh Chau Nguyen[1], Hosein Rostamian[2,3], Ana Raman[4], Pengcheng Wei [5,6], Dustin C. Becht [1], Annette H. Erbse[7], Brianna J. Klein[1], Tonya M. Gilbert[4], Gongyi Zhang [5], M. Andres Blanco[8], Brian D. Strahl [2], Sean D. Taverna[4,9] & Tatiana G. Kutateladze [1] ✉

The NuA3 complex is a major regulator of gene transcription and the cell cycle in yeast. Five core subunits are required for complex assembly and function, but it remains unclear how these subunits interact to form the complex. Here, we report that the Taf14 subunit of the NuA3 complex binds to two other subunits of the complex, Yng1 and Sas3, and describe the molecular mechanism by which the extra-terminal domain of Taf14 recognizes the conserved motif present in Yng1 and Sas3. Structural, biochemical, and mutational analyses show that two motifs are sandwiched between the two extra-terminal domains of Taf14. The head-to-toe dimeric complex enhances the DNA binding activity of Taf14, and the formation of the hetero-dimer involving the motifs of Yng1 and Sas3 is driven by sequence complementarity. In vivo assays in yeast demonstrate that the interactions of Taf14 with both Sas3 and Yng1 are required for proper function of the NuA3 complex in gene transcription and DNA repair. Our findings suggest a potential basis for the assembly of three core subunits of the NuA3 complex, Taf14, Yng1 and Sas3.

The histone acetyltransferase complex NuA3 (nucleosomal acetylation of H3) mediates responses to environmental stresses and is implicated in transcription and cell cycle regulation[1,2]. The canonical NuA3 complex acetylates histone H3, generating primarily the epigenetic mark H3K14ac[3], and consists of five core subunits: the catalytic subunit Sas3 (something about silencing 3) and the nonenzymatic subunits Nto1 (NuA three ORF1), Eaf6 (Esa1p associated factor 6), Yng1 (yeast inhibitor of growth protein 1) and Taf14 (transcription initiation factor subunit 14) (Fig. 1a). The Yng1 and Taf14 subunits contain histone binding modules, the so-called readers of PTMs in histone proteins[4,5].

The plant homeodomain (PHD) finger of Yng1 binds to (or reads) trimethylated lysine 4 of H3 (H3K4me3)[6–10], whereas the YEATS (Yaf9, ENL, AF9, Taf14, Sas5) domain of Taf14 recognizes acylated lysine 9 of H3 (H3K9acyl)[11–14] (Fig. 1b). Previous studies showed that the association of the NuA3 complex with chromatin depends in part on Yng1 and Taf14[1,6,15], however, little is known about the NuA3 complex assembly and how the complex subunits are linked and cooperate directing its enzymatic activity at specific genomic sites.

Genetic studies have shown that the interaction of Yng1 with H3K4me3 helps to recruit the NuA3 complex to promoters of actively

[1]Department of Pharmacology, University of Colorado School of Medicine, Aurora, CO 80045, USA. [2]Department of Biochemistry & Biophysics, The University of North Carolina School of Medicine, Chapel Hill, NC 27599, USA. [3]Curriculum in Genetics and Molecular Biology, The University of North Carolina School of Medicine, Chapel Hill, NC 27599, USA. [4]Department of Pharmacology and Molecular Sciences, Johns Hopkins University School of Medicine, Baltimore, MD 21205, USA. [5]Department of Biomedical Research, National Jewish Health, Denver, CO 80206, USA. [6]Guangxi Key Laboratory of Special Biomedicine, School of Medicine, Guangxi University, Nanning 530004, China. [7]Department of Biochemistry, University of Colorado, Boulder, CO 80303, USA. [8]Department of Biomedical Sciences, University of Pennsylvania, School of Veterinary Medicine, Philadelphia, PA 19104, USA. [9]Department of Biochemistry and Molecular Biology, University of Arkansas for Medical Sciences, Little Rock, AR 72202, USA. ✉e-mail: tatiana.kutateladze@cuanschutz.edu

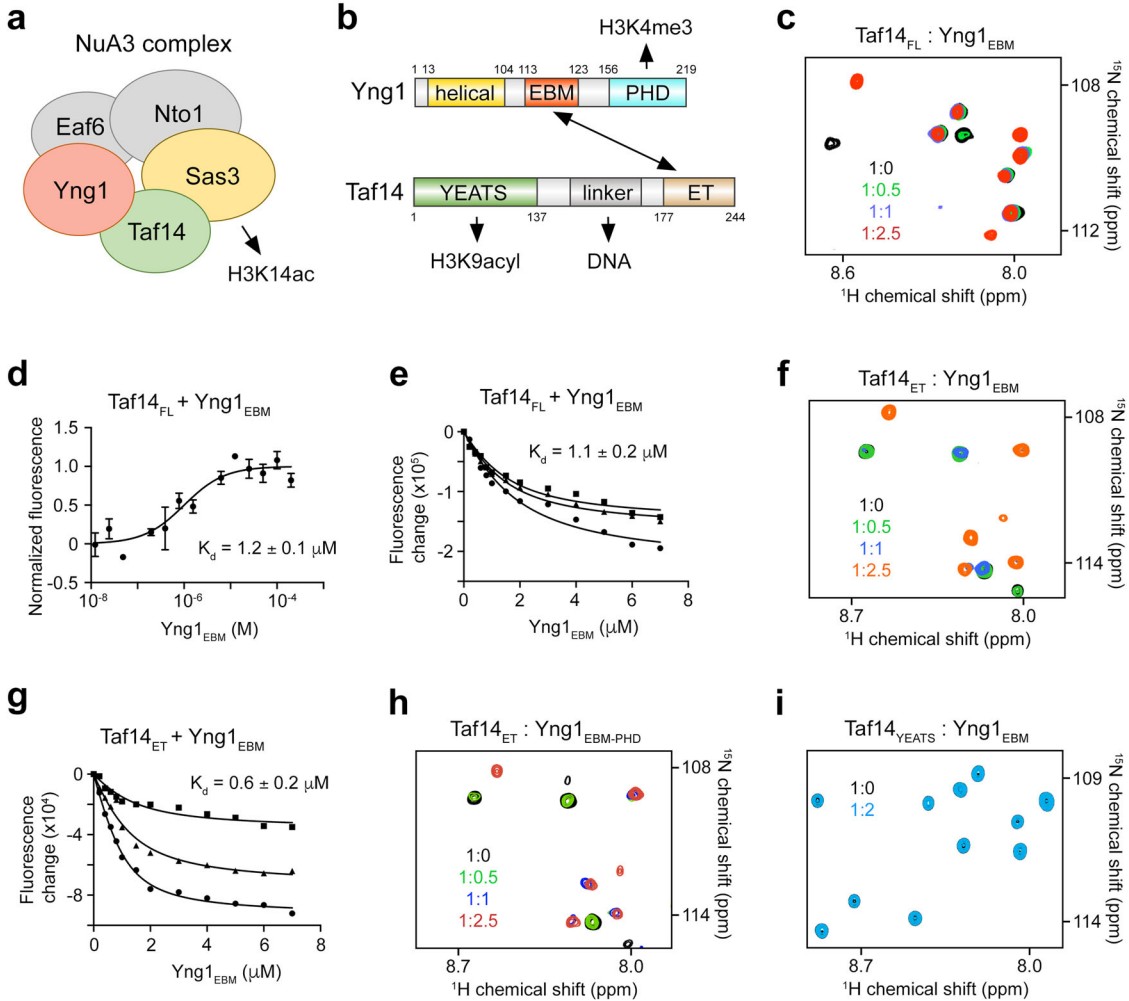

**Fig. 1 | Yng1 contains EBM, which is recognized by Taf14ET. a** Schematic of the yeast histone acetyltransferase NuA3 complex, which catalyzes acetylation of lysine 14 in histone H3 (H3K14ac). **b** Domain architecture of Taf14 and Yng1. H3K4me3, H3K9acyl and DNA, the known ligands for the PHD finger of Yng1, the YEATS domain of Taf14, and the linker of Taf14, respectively, are indicated. **c** Overlayed $^1H,^{15}N$ HSQC spectra of $Taf14_{FL}$ recorded in the presence of increasing amounts of $Yng1_{EBM}$ peptide. The spectra are color coded according to the protein:peptide molar ratio. **d** MST binding curve for the interaction of $Taf14_{FL}$ with $Yng1_{EBM}$. The $K_d$ value represents average of three independent measurements ± SEM, and error bars represent SEM for each point. $n = 3$. **e** Tryptophan fluorescence binding curves for the interaction of $Taf14_{FL}$ with $Yng1_{EBM}$. The $K_d$ value represents average of three

independent measurements, with error calculated as SD between the runs. **f** Superimposed $^1H,^{15}N$ HSQC spectra of $Taf14_{ET}$ recorded in the presence of increasing amount of $Yng1_{EBM}$. The spectra are color coded according to the protein:peptide molar ratio. **g** Tryptophan fluorescence binding curves for the interaction of $Taf14_{ET}$ with $Yng1_{EBM}$. The $K_d$ value represents average of three independent measurements, with error calculated as SD between the runs. **h, i** Superimposed $^1H,^{15}N$ HSQC spectra of $Taf14_{ET}$ (**h**) or $Taf14_{YEATS}$ (**i**) recorded in the presence of increasing amounts of $Yng1_{EBM-PHD}$ (**h**) or $Yng1_{EBM}$ peptide (**i**). The spectra are color coded according to the protein:ligand molar ratio. The apparent $K_d$ values in (**d, e** and **g**) were calculated using a 1:1 stoichiometry. Source data are provided as a Source Data file.

transcribed target genes, where this complex acetylates H3K14, facilitating transcription initiation[6]. Yng1 is a small 219-residue protein whose C-terminal PHD finger recognizes H3K4me3 through a canonical 'aromatic cage' dependent binding mechanism[6]. The Yng1 region N-terminal to the PHD finger remains uncharacterized, though AlphaFold modeling (Uniprot) predicts that it may adopt an α-helical fold, likely due to similarity with the N-terminal regions of homologous human inhibitor of growth proteins ING4 and ING5[16,17].

Taf14 is a core component of several chromatin associating complexes that regulate gene expression, DNA damage response and cytoskeletal organization in yeast cells. These include the general transcription factors complexes TFIID and TFIIF, the ATP-dependent chromatin-remodeling complexes INO80, SWI/SNF and RSC, and the acetyltransferase complex NuA3[1,18–23]. Along with the H3K9acyl-binding YEATS domain, Taf14 contains the C-terminal extra-terminal (ET) domain that associates with co-factors in RSC and TFIID

complexes, and a short region with DNA-binding activity that links the YEATS and ET domains, hence named a linker[24–26] (Fig. 1b).

Sas3 belongs to the MYST (MOZ, Ybf2/Sas3, Sas2, TIP60) family of histone acetyltransferases (HATs), distinguished by their catalytic MYST domain and is a yeast counterpart of the human HATs MOZ and MORF[27]. The central MYST domain of Sas3 is followed by a long stretch of aspartate and glutamate residues which are essential in the association with the transcriptional elongation factor Spt16[1]. Although the selectivity of the MYST domain of Sas3 toward acetylation of H3 and particularly H3K14 has been established in vitro and in vivo[1,3], Sas3 remains the least well understood MYST family member.

In this study, we show that the Taf14 subunit of the NuA3 complex interacts with EBMs from other two complex subunits, Yng1 and Sas3, and describe the molecular mechanism by which dimeric EBM is recognized by the ET domain of Taf14. Our findings demonstrate that both interactions of Taf14 with Sas3 and Yng1 are required for proper function of the NuA3 complex in gene transcription and DNA repair

and suggest a model for the assembly of the three core subunits of the NuA3 complex.

## Results and Discussion

### Yng1$_{EBM}$ is recognized by Taf14$_{ET}$

Taf14 has been shown to bind other subunits in the TFIID and RSC complexes through its ET domain[24,25], therefore we explored the idea that Taf14 has a similar function in the NuA3 complex. We analyzed amino acid sequences of the NuA3 core subunits and identified the LLLKI sequence of Yng1 (aa 116-120) as a potential ligand of Taf14. To determine whether Taf14 binds to Yng1, we carried out NMR, microscale thermophoresis (MST), and fluorescence experiments. For NMR, we produced full length uniformly $^{15}N$-labeled Taf14 (Taf14$_{FL}$) and recorded its $^1H,^{15}N$ heteronuclear single quantum coherence (HSQC) spectra. Highly dispersed amide resonances of Taf14$_{FL}$ indicated that the apo-state is folded. Titration of unlabeled Yng1 peptide (aa 113-124 of Yng1) into the Taf14$_{FL}$ sample led to substantial chemical shift perturbations (CSPs) (Fig. 1c and Supplementary Fig. 1). A number of resonances corresponding to the unbound state of Taf14$_{FL}$ decreased in intensity and disappeared, and concomitantly, another set of resonances that corresponds to the bound state of Taf14$_{FL}$ gradually appeared. This pattern of CSPs is characteristic of the slow exchange regime on the NMR time scale and indicates tight binding, which was confirmed by measuring binding affinity of Taf14$_{FL}$ to the Yng1 peptide. The apparent dissociation constant ($K_d$) determined by MST and tryptophan fluorescence was found to be 1.2 μM and 1.1 μM, respectively (Fig. 1d, e).

We next examined individual domains of Taf14, i.e. YEATS (Taf14$_{YEATS}$) and ET (Taf14$_{ET}$), by NMR to establish their roles in the interaction with Yng1. Gradual addition of the Yng1 peptide into $^{15}N$-labeled Taf14$_{ET}$ caused CSPs that were on par with CSPs observed in Taf14$_{FL}$, indicating that Taf14$_{ET}$ binds to this sequence of Yng1, from here on referred to as the ET-binding motif (Yng1$_{EBM}$) (Fig. 1f and Supplementary Fig. 1). However, a 0.6 μM binding affinity of Taf14$_{ET}$ to Yng1$_{EBM}$ implied a tighter binding compared to the binding of Taf14$_{FL}$ to Yng1$_{EBM}$ (Fig. 1g). These data support the previous observations that while an isolated Taf14$_{ET}$ is more freely available for the interactions with ligands, it is less freely available in the context of the full-length protein due to inhibitory contacts with the linker region of Taf14 (Fig. 1e)[25,26].

Full length Yng1 expressed in *E. coli* formed inclusion bodies, however we were able to produce a soluble construct containing EBM coupled to the PHD finger of Yng1 (aa 100-219 of Yng1, Yng1$_{EBM-PHD}$) and tested it by NMR. Because similar patterns of CSPs were observed in $^1H,^{15}N$ HSQC spectra of Taf14$_{ET}$ upon titration with either Yng1$_{EBM-PHD}$ or Yng1$_{EBM}$, we concluded that the PHD finger of Yng1 does not affect binding of Taf14$_{ET}$ to Yng1$_{EBM}$ (Fig. 1h and Supplementary Fig. 2). No CSPs were induced in $^{15}N$-labeled Taf14$_{YEATS}$ by Yng1$_{EBM}$, implying that Taf14$_{YEATS}$ is also not involved in this interaction (Fig. 1i and Supplementary Fig. 2).

### Structural basis for the Taf14-Yng1 complex formation

To gain insight into the binding mechanism, we co-crystallized Taf14$_{ET}$ and Yng1$_{EBM}$ and determined the structure of the Taf14$_{ET}$-Yng1$_{EBM}$ complex refining it to a 1.9 Å resolution (Fig. 2 and Supplementary Table 1). In the crystal structure, Taf14$_{ET}$ forms a head-to-toe dimer, which creates a deep, elongated binding groove for dimeric Yng1$_{EBM}$, and mass photometry data indicate that the Taf14$_{FL}$-Yng1$_{EBM}$ complex can also form a dimer in solution (Supplementary Fig. 3). Each Taf14$_{ET}$ monomer, colored light blue and wheat, folds into a three-helix bundle and a two-stranded anti-parallel β-sheet linking helices α2 and α3 (Fig. 2a and Supplementary Fig. 4). The α1 helices of both Taf14$_{ET}$ monomers (α1 and α1') are packed against each other in an antiparallel manner. The α1-α1' dimerization interface of Taf14$_{ET}$ is stabilized through van der Waals and hydrophobic contacts. The side chains of

the symmetry-related F184 residues in α1 and α1' are buried in the dimer interface, and although the aromatic rings are not ideally parallel to produce π stacking, they are only ~4.5 Å apart.

Dimeric Yng1$_{EBM}$ forms a long two-stranded anti-parallel β-sheet (one β strand is colored yellow, and another is red), which pairs with the β2 and β2' strands from Taf14$_{ET}$ monomers and is restrained through extensive intermolecular interactions (Fig. 2b, c and Supplementary Fig. 4). Symmetric β-sheet contacts are observed between the backbone amides of L122, I120 and L118 of each Yng1$_{EBM}$ strand and G218, F220 and I222 of the respective β2 and β2' strands of Taf14$_{ET}$ (Fig. 2c). Additional hydrogen bonds and electrostatic interactions, engaging residues of the Taf14$_{ET}$ monomer 1 (light blue) and Yng1$_{EBM}$ (yellow), further stabilize the complex. The side chain amino groups of K115, K119 and K124 of Yng1$_{EBM}$ form salt bridges with the carboxyl groups of D223, E219 and E191 of Taf14$_{ET}$, and the side chain amino group of K123 of Yng1$_{EBM}$ shares a hydrogen bond with the backbone carbonyl group of E216 of Taf14$_{ET}$. Another hydrogen bond is formed between the backbone amide of L116 of Yng1$_{EBM}$ and the side chain hydroxyl group of Y225 of Taf14$_{ET}$. A distinct hydrogen bonding network links the Taf14$_{ET}$ monomer 2 (wheat) and Yng1$_{EBM}$ (red): the hydroxyl group of Y225 of Taf14$_{ET}$ is restrained by the backbone amide of L116 of Yng1$_{EBM}$, and the side chain amide of N121 of Yng1$_{EBM}$ is engaged with the carboxyl group of E219 of Taf14$_{ET}$. Nine residues in each Yng1$_{EBM}$ strand, from E113 to N121, form a long β-sheet, allowing for a large number of hydrophobic, electrostatic and polar contacts within the Taf14-Yng1 complex (Fig. 2d).

### Critical role of hydrophobic contacts

The Yng1$_{EBM}$-binding groove of the Taf14$_{ET}$ dimer is large and hydrophobic, and two molecules of Taf14$_{ET}$ almost encircle the dimeric Yng1$_{EBM}$ (Fig. 3a). The side chains of L117, L118, I120 and L122 from both strands of Yng1$_{EBM}$ are oriented toward and essentially buried in the hydrophobic groove. To assess the importance of the hydrophobic interface residues in the complex formation, we replaced L116, L118 and I120 of Yng1$_{EBM}$ individually with aspartate, as well as F220, I222 and L224 of Taf14$_{ET}$ individually with arginine and tested the mutated proteins and peptides by NMR (Fig. 3b–g and Supplementary Fig. 5). An almost negligible CSPs in $^1H,^{15}N$ HSQC spectra of wild type Taf14$_{ET}$ upon addition of I120D Yng1$_{EBM}$ indicated that I120 is required for this interaction, and CSPs induced by L118D Yng1$_{EBM}$ or L116D Yng1$_{EBM}$ revealed weak interactions characterized by $K_d$s of 430 μM and 180 μM, respectively (Fig. 3g). Such a considerable decrease in binding activity of Taf14$_{ET}$ suggested that the hydrophobic contacts are the driving force for this complex formation. Mutation of either F220, I222 or L224 in Taf14$_{ET}$ caused protein unfolding, pointing to a critical role of these hydrophobic residues in structural stability of Taf14.

We note, that human ING4 or homologous ING5 is dimeric[16,17], and either is present in the human HAT complexes as a native subunit[28]. Despite the high degree similarity to Yng1, homologous Yng2, a well-known component of the yeast NuA4 complex, has not been detected in the yeast NuA3 complex. A lack of EBM in the Yng2 sequence may account for the inability of Yng2 to interact with Taf14, therefore precluding Yng2 to be part of the NuA3 complex (Fig. 3h).

### Yng1$_{EBM}$ enhances DNA binding activity of Taf14

As Taf14 was found to bind DNA via its linker region[25], we tested whether the interaction with Yng1$_{EBM}$ impacts the DNA binding function of Taf14$_{FL}$ using electrophoretic mobility shift assays (EMSA). We mixed increasing amounts of Taf14$_{FL}$ with 601 Widom DNA without and with Yng1$_{EBM}$ and resolved the reaction mixtures on a 8% native polyacrylamide gel. As shown in Fig. 4a and Supplementary Fig. 6, Taf14$_{FL}$ shifts the free 601 DNA band, indicating formation of the Taf14$_{FL}$-DNA complex. The presence of Yng1$_{EBM}$ enhanced binding of Taf14$_{FL}$ to DNA, supporting previous observations that the DNA binding activity of Taf14 is regulated by Taf14$_{ET}$[25]. Because of its

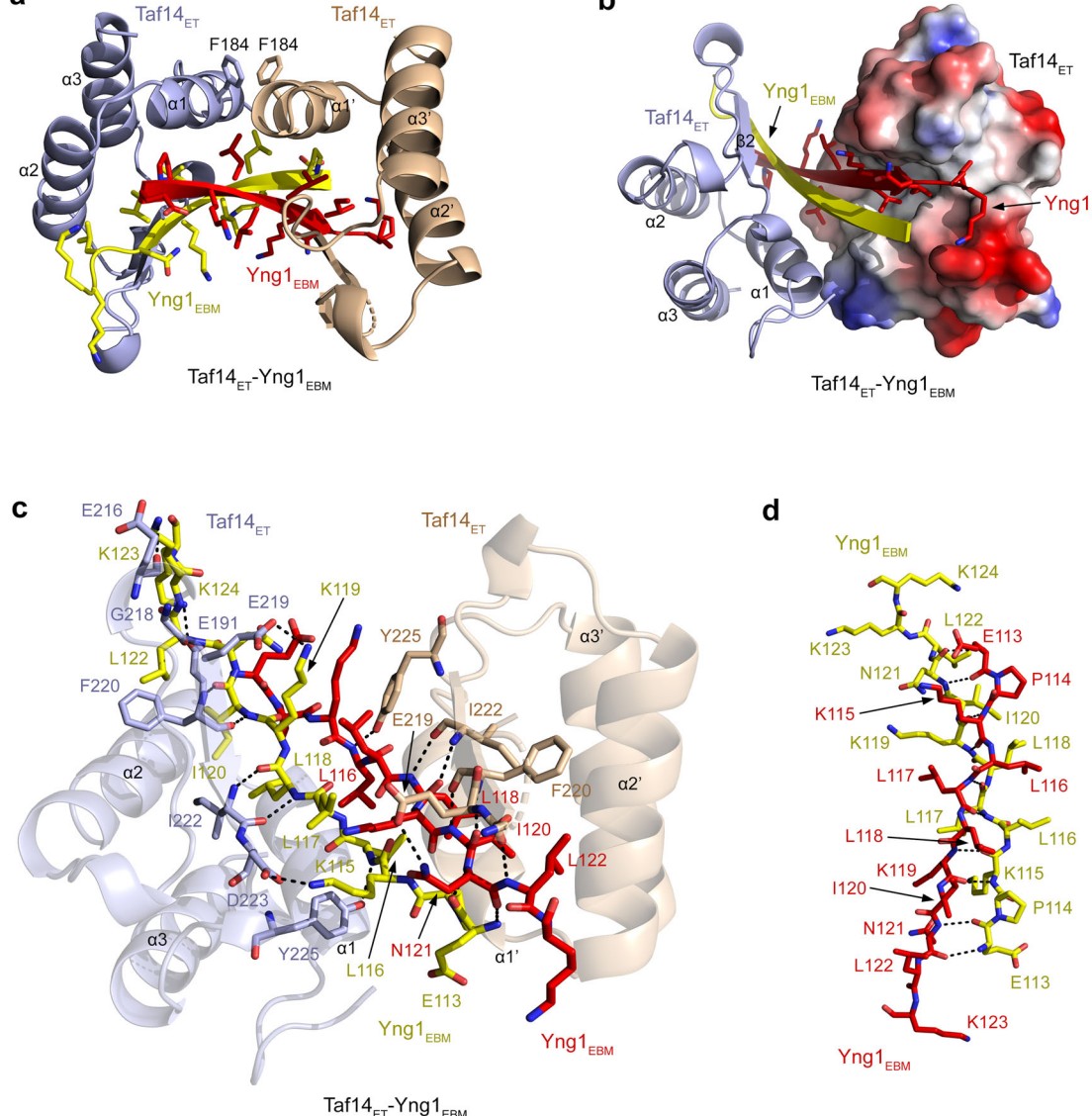

**Fig. 2 | Structural mechanism for the Taf14ET-Yng1EBM complex formation.**
**a** Ribbon diagram of the Taf14$_{ET}$-Yng1$_{EBM}$ complex structure. Dimeric Taf14$_{ET}$ is colored light blue (monomer 1) and wheat (monomer 2), and two monomers of dimeric Yng1$_{EBM}$ are colored yellow and red. **b** The Taf14$_{ET}$-Yng1$_{EBM}$ complex with monomer 2 depicted as a surface model colored according to its electrostatic potential (positive potential, blue; negative potential, red). **c** Ribbon diagram of the Taf14$_{ET}$-Yng1$_{EBM}$ complex with two molecules of Yng1$_{EBM}$ shown as sticks. Residues involved in the interaction between Taf14$_{ET}$ and Yng1$_{EBM}$ are labeled. Dashed lines represent hydrogen bonds. **d** Residues in two Yng1$_{EBM}$ are shown as sticks and labeled. Dashed lines represent hydrogen bonds and salt bridges.

association with Taf14$_{ET}$, the linker is not freely accessible in the context of full-length protein but is released for DNA binding once Taf14$_{ET}$ is ligand-bound[25]. Our data indicate that Yng1$_{EBM}$ outcompetes the linker for binding to Taf14$_{ET}$. Unexpectedly, Yng1$_{EBM}$ increased the DNA binding activity of the isolated Taf14$_{ET}$ as well (Fig. 4b and Supplementary Fig. 6). This increase could be due to the extended positively charged patch formed by the dimeric Taf14$_{ET}$-Yng1$_{EBM}$ complex on the side opposite to the Yng1$_{EBM}$-binding groove (Fig. 4c, outlined by green dots). We then examined whether the interactions of Taf14$_{ET}$ with Yng1$_{EBM}$ and of Taf14$_{YEATS}$ with crotonylated H3K9 (H3K9cr) affect each other. The binding affinity of Taf14$_{FL}$ to Yng1$_{EBM}$ in the presence of H3K9cr (K$_d$ of 0.9 µM) indicated that the two functions of Taf14$_{FL}$ are independent (Fig. 4d). Together, these results suggest that the formation of the dimeric complex of Taf14$_{FL}$ can bridge two EBM-containing proteins in an antiparallel manner, enhances DNA binding activity of Taf14$_{FL}$, and does not impede binding of Taf14$_{YEATS}$ to H3K9cr (Fig. 4e).

## Taf14, Yng1 and Sas3 localize to gene promoters
To explore the functional cooperation of Taf14 and Yng1, we analyzed ChIP-exo data from *Saccharomyces cerevisiae* TAP-tagged strains in BY4741 background previously reported for these proteins[29]. We found that Yng1 occupies promoters of 71 genes, and remarkably 70 of these gene promoters were co-bound by Taf14 (Fig. 4f). These Yng1-Taf14 co-bound genes were markedly enriched for ribosomal functionality and regulators of translation (Fig. 4g and Supplementary Data 1), suggesting that Taf14 and Yng1 co-regulate aspects of these biological processes. Further analysis of genomic occupancies revealed that Taf14, Yng1, and the catalytic subunit of the NuA3 complex, Sas3, each bound nearly exclusively to promoters, with their binding sites mapping to 1096, 71, and 9 unique genes, respectively (Fig. 4h). Furthermore, 6 gene promoters bound by Sas3 were also co-bound by Yng1 and Taf14, suggesting that Sas3, Yng1 and Taf14 cooperate in transcriptional regulation in vivo.

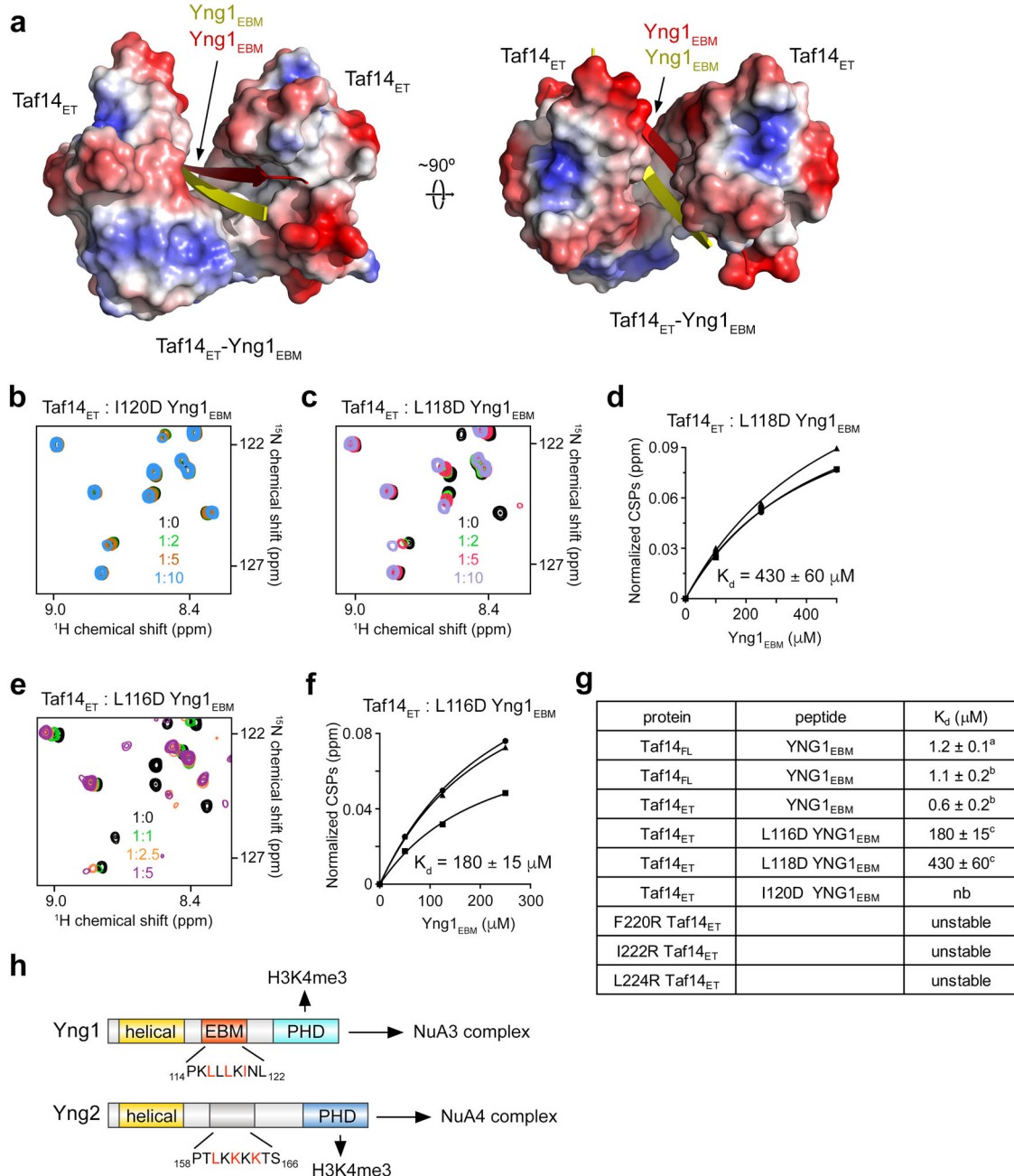

**Fig. 3 | Role of hydrophobic contacts. a** Electrostatic surface potential of the Taf14$_{ET}$-Yng1$_{EBM}$ complex, with blue and red colors representing positive and negative charges, respectively. Dimeric Yng1$_{EBM}$ is shown as yellow and red β strands. (b-f) NMR assays to monitor interactions of Taf14$_{ET}$ with the indicated mutants of Yng1$_{EBM}$. Overlayed $^1$H,$^{15}$N HSQC spectra of Taf14$_{ET}$ recorded in the presence of increasing amounts of the mutated Yng1$_{EBM}$ peptides are shown in (**b**, **c**, and **e**). The spectra are color coded according to the protein:peptide molar ratio. Binding curves used to determine K$_d$s by NMR are shown in (**d**, **f**). The K$_d$ value represents average of CSPs observed in three amides ± SD. **g** Binding affinities of Taf14 proteins to the indicated ligands measured by [a]MST, [b]fluorescence, and [c]NMR. nb, no binding (**h**) Domain architecture of Yng1, a subunit of the NuA3 complex, and of Yng2, a subunit of the NuA4 complex. EBM sequence in Yng1 and the corresponding sequence in Yng2 are shown. The apparent K$_d$ values in (**d**, **f**, and **g**) were calculated using a 1:1 stoichiometry. Source data are provided as a Source Data file.

## Sas3$_{EBM}$ is recognized by Taf14$_{ET}$

The catalytic subunit of the NuA3 complex, Sas3, contains the LKVRI sequence (aa 114-118) that might also serve as EBM for Taf14[24] (Fig. 5a). Indeed, Taf14$_{FL}$ associated with the Sas3 peptide (aa 111-122, Sas3$_{EBM}$) with the binding affinity of 21 µM as measured by tryptophan fluorescence (Fig. 5b). The interaction between Taf14$_{ET}$ and Sas3$_{EBM}$ was further confirmed by NMR. Sas3$_{EBM}$ caused large CSPs in $^1$H,$^{15}$N HSQC spectra of $^{15}$N-labeled Taf14$_{ET}$, and the K$_d$ value for the Taf14$_{ET}$-Sas3$_{EBM}$ interaction was found to be 18 µM (Fig. 5c, d and Supplementary Fig. 7).

We note that although the binding affinity of Taf14$_{FL}$ or Taf14$_{ET}$ to Sas3$_{EBM}$ was moderately weaker than that of toward Yng1$_{EBM}$, it was still in the range of typical affinities exhibited by chromatin associating proteins. Much like hydrophobic contacts in the Taf14-Yng1 complex, the hydrophobic contacts were required for the formation of the Taf14-Sas3 complex. The single point substitutions V116D and I118D in Sas3$_{EBM}$ abrogated binding of Taf14$_{ET}$, and L114D mutation in Sas3$_{EBM}$ substantially reduced this interaction (K$_d$ = 540 µM) (Fig. 5e–h and Supplementary Fig. 7).

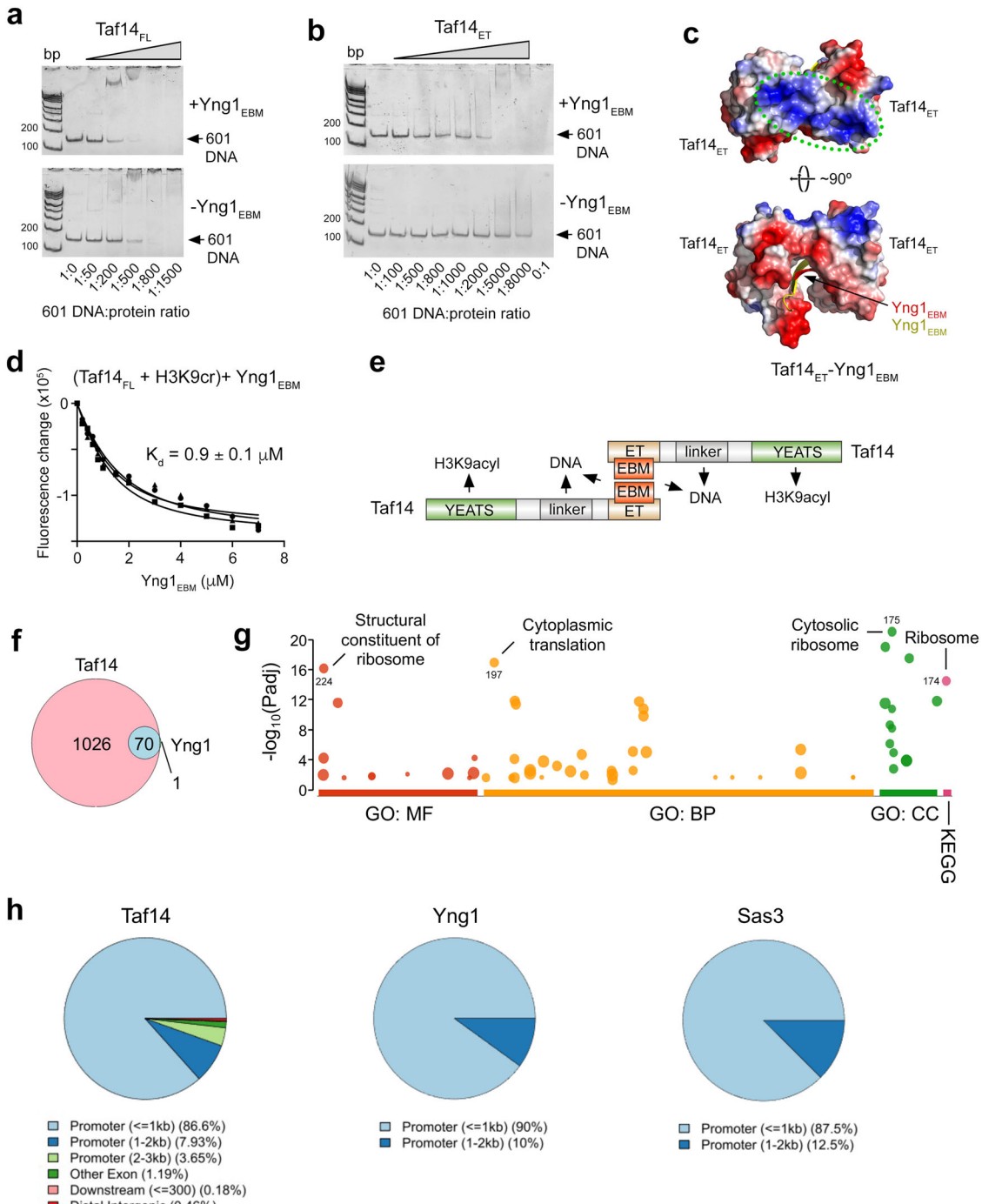

**Fig. 4 | Yng1EBM enhances DNA binding activity of Taf14. a, b** EMSAs of 601 DNA in the presence of increasing amounts of Taf14$_{FL}$ (**a**) or Taf14$_{ET}$ (**b**) with or without Yng1$_{EBM}$. DNA:protein ratio is shown below gel images. Experiments in (**a, b**) were performed once and twice, respectively, see also Supplementary Fig. 6. **c** Rotated views of the Taf14$_{ET}$-Yng1$_{EBM}$ complex shown in Fig. 3a and colored according to its electrostatic potential (positive potential, blue; negative potential, red). An extended positively charged patch on the surface is outlined by green oval. **d** Binding curves used to determine apparent K$_d$ for the interaction of Taf14$_{FL}$ with Yng1$_{EBM}$ in the presence of H3K9cr peptide at a 1:10 Taf14$_{FL}$:H3K9cr molar ratio by tryptophan fluorescence (using a 1:1 stoichiometry). The apparent K$_d$ value represents average of three independent measurements, with error calculated as SD between the runs. **e** A model for the formation of the complex between two molecules of Taf14 and two proteins containing EBM. H3K9acyl and DNA, the known ligands of the YEATS domain of Taf14, and the linker of Taf14, respectively, are indicated. **f** Overlap in genes with promoters bound by Taf14 and Yng1. **g** GO and KEGG category enrichments of genes with promoters co-occupied by Taf14 and Yng1. Circle size represents the approximate number of genes in the enrichment pathway. The exact number of genes is shown for the top pathway in each GO/KEGG category. **h** Genomic distribution of Taf14, Yng1, and Sas3 binding sites. Source data are provided as a Source Data file.

The finding that two Taf14$_{ET}$ bring together two EBMs suggests a possibility for the formation of the hetero-dimeric complex containing one EBM from Yng1 and another EBM from Sas3 (Fig. 6a). Notably, amino acid sequences of Yng1$_{EBM}$ and Sas3$_{EBM}$ have a high degree of charge-charge and hydrophobicity complementarity when aligned in an antiparallel manner (Fig. 6b). The AlphaFold2-predicted models of the Taf14$_{ET}$-Yng1$_{EBM}$ and Taf14$_{ET}$-Sas3$_{EBM}$ complexes suggest that, in the hetero-tetrameric complex, Sas3$_{EBM}$ can occupy the same position

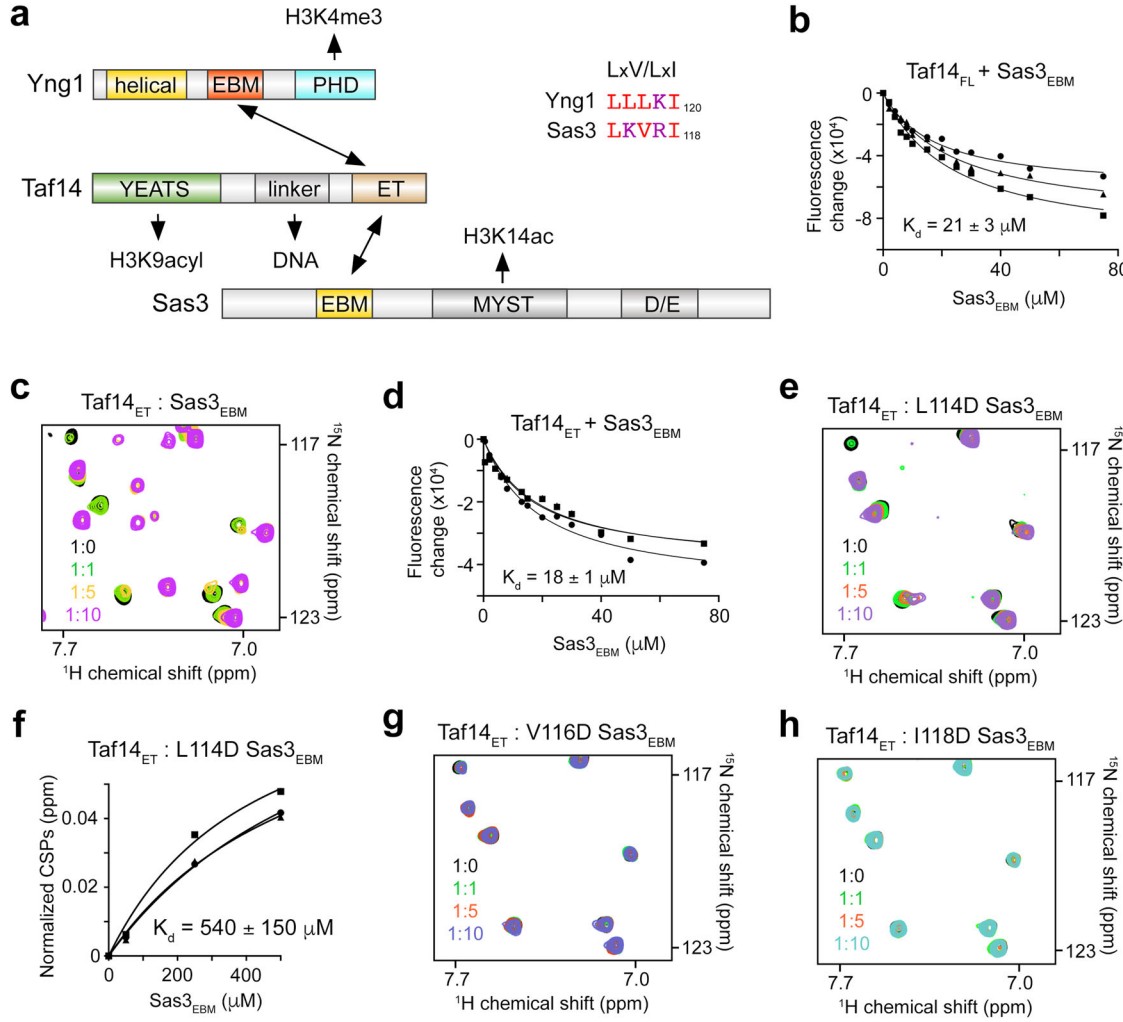

**Fig. 5 | Sas3 also contains EBM, which is recognized by Taf14ET. a** Domain architecture of Yng1, Taf14 and Sas3. Sequence alignment of Yng1$_{EBM}$ and Sas3$_{EBM}$ is shown on the right. **b** Binding curves used to determine $K_d$ for the interaction of Taf14$_{FL}$ with Sas3$_{EBM}$ by tryptophan fluorescence. The $K_d$ value represents average of three independent measurements, with error calculated as SD between the runs. **c** Superimposed $^1$H,$^{15}$N HSQC spectra of Taf14$_{ET}$ recorded in the presence of increasing amounts of Sas3$_{EBM}$ peptide. The spectra are color coded according to the protein:peptide molar ratio. **d** Binding curves used to determine $K_d$ for the interaction of Taf14$_{ET}$ with Sas3$_{EBM}$ by tryptophan fluorescence. The $K_d$ value represents average of three independent measurements, with error calculated as SD between the runs. **e–h** NMR assays to monitor interactions of Taf14$_{ET}$ with the indicated mutants of Sas3$_{EBM}$. Overlayed $^1$H,$^{15}$N HSQC spectra of Taf14$_{ET}$ recorded in the presence of increasing amounts of the mutated Sas3$_{EBM}$ peptides are shown in (**e, g, h**). The spectra are color coded according to the protein:peptide molar ratio. Binding curves used to determine $K_d$ by NMR are shown in (**f**). The $K_d$ value represents average of CSPs observed in three amides ± SD. The apparent $K_d$ values in (**b, d** and **f**) were calculated using a 1:1 stoichiometry. Source data are provided as a Source Data file.

as one of Yng1$_{EBM}$ in the crystal structure of the dimeric Taf14$_{ET}$-Yng1$_{EBM}$ complex (Fig. 6c). Although additional experimental studies are needed to confirm this arrangement, circular dichroism (CD) spectra show that the mixture of Yng1$_{EBM}$ and Sas3$_{EBM}$ in a 1:1 ratio stabilizes the structure of the complex with Taf14$_{ET}$ to the same extent as individual Yng1$_{EBM}$ and Sas3$_{EBM}$ stabilize it (Supplementary Fig. 8). Furthermore, the DNA binding activity of Taf14$_{ET}$ was augmented by either the mixture of EBMs or Yng1$_{EBM}$ in EMSA assays (Supplementary Fig. 9 and Fig. 4b), and comparable molecular mass of Taf14$_{FL}$ was measured in solution in the presence of Yng1$_{EBM}$ or the mixture of EBMs by mass photometry (Supplementary Fig. 3).

### Binding of Taf14 to Yng1$_{EBM}$ and Sas3$_{EBM}$ is required for the functional NuA3 complex

To evaluate the biological consequences of the Taf14-Sas3 and Taf14-Yng1 interactions within the NuA3 complex, we generated *Saccharomyces cerevisiae* TAP-tagged strains in BY4741 background harboring the mutations I118D in Sas3 and I120D in Yng1 that disrupt binding of

Taf14. The WT and modified cells were then plated on media containing various compounds or growth conditions to discern drug and growth sensitivity phenotypes associated with the loss of binding of Taf14 to Yng1 or Sas3 in the NuA3 complex (Fig. 6d, e and Supplementary Fig. 10). Given the role of NuA3 in transcription elongation[30], we first investigated the impact of elimination of the direct Taf14-Sas3 and Taf14-Yng1 contacts on sensitivity to the transcription elongation inhibitor 6-azauracil (6-AU). As shown in Fig. 6d, strains containing point mutations in either Sas3 or Yng1 that prevent Taf14 association exhibited strong sensitivity to 6-AU, with a near loss of growth observed for the Sas3 I118D mutation. These data reveal that both interactions of Taf14 with Sas3 and Yng1 are required for the proper activity of the NuA3 complex in gene transcription, and elimination of either abrogates this function. The same trend albeit less pronounced was observed under genotoxic agents that induce DNA damage and/or replication stress, such as hydroxyurea (HU) and methyl methanesulfonate (MMS). In these conditions, the strain containing the Sas3 I118D mutation was more sensitive than the strain containing the Yng1

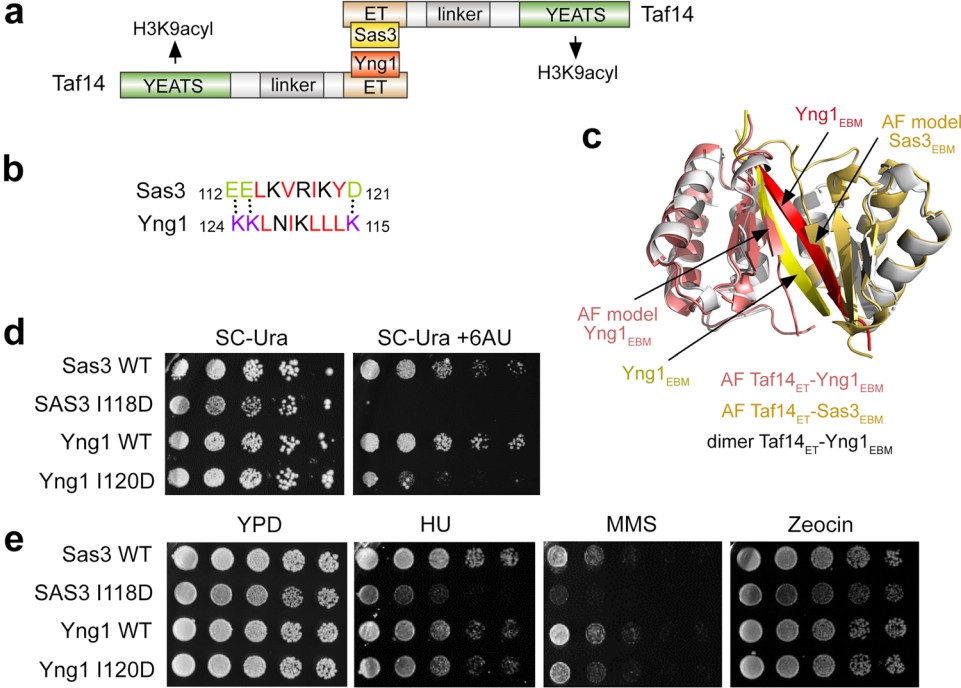

**Fig. 6 | Binding of Taf14 to Yng1EBM and Sas3EBM is required for the functional NuA3 complex. a** A model for the formation of the complex between two molecules of Taf14 and two EBMs, one EBM from Yng1 and another EBM from Sas3. The ligand of Taf14$_{YEATS}$, H3K9acyl, is indicated. **b** Sequence alignment of Yng1$_{EBM}$ and Sas3$_{EBM}$ presented in an antiparallel manner. Dotted lines represent complementary charge-charge contacts. **c** Overlay of the AlphaFold2-predicted models of the Taf14$_{ET}$-Yng1$_{EBM}$ (salmon) and Taf14$_{ET}$-Sas3$_{EBM}$ (gold) complexes with the structure of Taf14$_{ET}$-Yng1$_{EBM}$ (dimeric Yng1$_{EBM}$ is colored red and yellow as in Fig. 2). **d** 0.5 OD$_{600}$ of the indicated yeast strains were 5-fold serially diluted on synthetic complete (SC) medium lacking Uracil (SC -Ura) or SC -Ura plates supplemented with 100 μg/ml 6-azauracil (6-AU) and grown for 4 days. **e** 0.5 OD$_{600}$ of the indicated yeast strains were 5-fold serially diluted onto YPD-containing plates supplemented with either 100 mM hydroxyurea (HU), 0.03% methanesulfonate (MMS), or 5 μg/ml Zeocin and grown at 30 °C for 3 days.

I120D mutation. Interestingly, neither mutant was sensitive to Zeocin but Sas3 I118D was temperature sensitive and showed a severe growth defect at both 37 °C and 16 °C (Supplementary Fig. 10). These phenotypes underscore the importance of the Sas3-Taf14 and Yng1-Taf14 interactions in the biological activities of the NuA3 complex and suggest that the NuA3 complex is essential in maintaining the integrity of DNA replication and DNA repair processes within the cell.

In conclusion, this work underscores the ability of the Taf14 subunit of the NuA3 complex to interact with EBMs from other two complex subunits, Yng1 and Sas3, and demonstrates that both interactions are required for the transcriptional and DNA repair activities of the NuA3 complex. Our structural and functional analyses suggest a model in which the Taf14 subunit may bridge Yng1 and Sas3 through the formation of a heterodimeric assembly. Future cryo-EM structural studies will be important to fully evaluate this model and determine the mechanisms by which Sas3, Yng1 and Taf14 are engaged with other two core subunits of the NuA3 complex and with chromatin.

## Methods

### Protein expression and purification

The *S. cerevisiae* Taf14$_{FL}$ (aa 1-244) (UniProt code P35189), Taf14$_{YEATS}$ (aa 1-137), Taf14$_{ET}$ (aa 168-243) and Yng1$_{EBM-PHD}$ (aa 100-219) (UniProt code Q08465) were cloned into pGEX-6P1, pET22b or pCIOX vectors. The F220R, I222R and L224R mutants of Taf14$_{ET}$ were generated using QuikChange II site directed mutagenesis kit (Agilent). Sequences were confirmed by DNA sequencing. All proteins were expressed in *Escherichia coli* Rosetta-2 (DE3) pLysS cells grown in either Terrific Broth, Luria Broth or in minimal media supplemented with $^{15}NH_4Cl$ (Sigma). Protein production was induced with 0.3 mM IPTG for 18 h at 16 °C. Bacteria were harvested by centrifugation and lysed by sonication. GST-fusion proteins were purified on glutathione agarose 4B beads (Thermo Fisher Sci). The GST-tag was cleaved with either

PreScission or tobacco etch virus (TEV) protease. After cleavage, Taf14$_{FL}$ and Taf14$_{YEATS}$ retained the pGEX-6P1vector-derived residues Gly and Pro. His-fusion proteins were incubated with Ni-NTA resin (ThermoFisher), washed and eluted with a gradient of imidazole. When necessary, removal of the His-tag was completed during an overnight dialysis, while incubating the protein with ULP1 protease. Proteins were further purified by size exclusion chromatography (SEC) and concentrated in Millipore concentrators (Millipore).

### MST

Microscale thermophoresis (MST) experiments were performed using a Monolith NT.115 instrument (NanoTemper). All experiments were performed using SEC purified His-Taf14$_{ET}$ and His-Taf14$_{FL}$ in a buffer containing 20 mM Tris-HCl (pH 7.5), 150 mM NaCl and 2.5 mM DTT or 0-0.2 mM TCEP as described. To monitor binding between Taf14$_{ET}$ or Taf14$_{FL}$ and Yng1$_{EBM}$ peptide Ac-EPKLLLKINLKK-NH2 (Synpeptide), His-tagged proteins were labeled using a His-Tag Labeling Kit RED-tris-NTA (2$^{nd}$ Generation, Nanotemper), and the concentration of each labeled protein was kept at 10 nM. Dissociation constants were determined using a direct binding assay, in which increasing amounts of unlabeled Yng1$_{EBM}$ were added stepwise. Protein concentration was calculated by measuring the UV absorbance at 280 nm, and peptide concentration was calculated based on the weight of the lyophilized powder. The measurements were performed at 40% LED and medium MST power with 3 s steady state, up to 20 s laser on time and 1 s off time. The apparent $K_d$ values were calculated using MO Affinity Analysis software (NanoTemper) using a 1:1 stoichiometry. Figures were generated in GraphPad PRISM.

### Fluorescence spectroscopy

Spectra were recorded at 25 °C on a Fluoromax-3 spectrofluorometer (HORIBA). The samples containing 1 μM Taf14$_{ET}$ or Taf14$_{FL}$ in 50 mM

Tris-HCl (pH 6.8), 150 mM NaCl and 1 mM DTT buffer or 50 mM Tris-HCl (pH 7.5), 150 mM NaCl and 0.2 mM TCEP buffer and progressively increasing concentrations of Yng1$_{EBM}$ Ac-EPKLLLKINLKK-NH2 or Sas3$_{EBM}$ Ac-SEELKVRIKYDS-NH2 peptides (Synpeptide) were excited at 295 nm. Emission spectra were recorded between 330 and 360 nm with a 0.5 nm step size and a 1 s integration time. The apparent $K_d$ values were determined using a nonlinear least-squares analysis and a 1:1 stoichiometry as described[31] and the equation:

$$\Delta I = \Delta I \max \left( ([L]+[P]+Kd) - \sqrt{([L]+[P]+Kd)^2 - 4[P][L]} \right)/2[P] \quad (1)$$

where [L] is concentration of the peptide, [P] is concentration of the protein, $\Delta I$ is the observed change of signal intensity and $\Delta I_{max}$ is the difference in signal intensity of the free and bound states of the protein. $K_d$ values were averaged over three separate experiments, and error was calculated as the standard deviation between the runs.

## NMR experiments

Nuclear magnetic resonance experiments were performed at 298 K on Varian INOVA 900 MHz (for Taf14$_{FL}$) and 600 MHz (for Taf14$_{ET}$ and Taf14$_{YEATS}$) spectrometers equipped with cryogenic probes as described[32]. The NMR samples contained 0.05–0.2 mM uniformly $^{15}$N-labeled Taf14 proteins in 50 mM Tris-HCl (pH 6.8–7.5), 150 mM NaCl, 0–5 mM DTT or 1x PBS (pH 6.6–7.5), supplemented with up to an additional 100 mM NaCl, 0–5 mM DTT or 0.2 mM TCEP and 8-10% D$_2$O. Binding was characterized by monitoring chemical shift changes in $^1$H,$^{15}$N HSQC spectra of the proteins induced by the addition of wild type or mutated Yng1$_{EBM}$ (aa 113-124), Sas3$_{EBM}$ (aa 111-122) and Yng1$_{EBM-PHD}$ (aa 100-219). NMR data were processed and analyzed using NMRPipe as previously described[32], with the apparent $K_d$s being determined by a nonlinear least-squares analysis and a 1:1 stoichiometry in GraphPad Prism using the equation:

$$\Delta\delta = \Delta\delta_{max} \left( ([L]+[P]+K_d) - \sqrt{([L]+[P]+K_d)^2 - 4[P][L]} \right)/2[P] \quad (2)$$

Where [L] is the concentration of the peptide, [P] is concentration of the protein, $\Delta\delta$ is the observed chemical shift change, and $\Delta\delta_{max}$ is the normalized chemical shift change at saturation. Normalized chemical shift changes were calculated using the equation $\delta = \sqrt{(\Delta\delta H)^2 + (\Delta\delta N/5)^2}$ (3), where $\Delta\delta$ is the change in chemical shift in parts per million (ppm).

## X-ray crystallography

Taf14$_{ET}$ was concentrated to ~2 mM in 50 mM Tris (pH 7.5), 150 mM NaCl. The protein was incubated with Yng1$_{EBM}$ peptide (Synpeptide) at a molar ratio of 1:2.5, for 16 h at 4 °C. The protein-peptide mixture was concentrated to 10 mg/ml. Crystals were grown using sitting-drop vapor diffusion method at 23 °C by mixing 0.3 μL of protein with 0.3 μL of well solution composed of 40% PEG 400, 0.1 M Tris pH 8.5, 0.2 M LiSO$_4$. X-ray diffraction data were collected on a beamline 4.2.2 at Advance Light Source. The diffraction data were processed using the iMosflm program 6[33], then scaled by Aimless Pointless in the CCP4 software suite 7[34]. The structure was determined by molecular replacement using the program PHASER[35], the Taf14$_{ET}$ structure (PDB ID: 7UHE) was used as a search model. The models from the molecular replacement were built using Crystallographic Object-Oriented Toolkit program 9[36] and subsequently subjected to refinement using PHENIX software[37]. Crystallography diffraction data collection and refinement statistics are summarized in Supplementary Table 1.

## DNA purification and preparation

Double stranded DNA containing the 147 bp 601 Widom DNA sequence in the pJ201 plasmid (32x) was transformed into DH5α cells. The plasmids containing the 601 sequence were amplified in LB media and extracted using the QIAprep® Spin Miniprep kit (QIAGEN). Amplification of double stranded 601 Widom DNA was performed by polymerase chain reaction (PCR) using Q5® High-Fidelity 2X master mix (New England Biolab). PCR product was verified by agarose gel and cleaned up by MinElute® Gel Extraction Kit (QIAGEN).

## EMSA

EMSAs were performed by mixing increasing amounts of Taf14 proteins with 0.25 pmol/lane of 601 DNA in 25 mM Tris-HCl pH 7.5, 25-100 mM NaCl, 0.2 mM ethylenediaminetetraacetic acid (EDTA), 10% glycerol and 1 mM DTT in 10 μL reaction volume. The Taf14 proteins were either preincubated with Yng1$_{EBM}$ at a 1:2.5 molar ratio or with Yng1$_{EBM}$ and Sas3$_{EBM}$ at a 1:1:1 molar ratio for a minimum of 1 h on ice or used without peptides prior to addition of DNA. Reaction mixtures were incubated at 4 °C for 10 min (2-2.5 μl of loading dye was added to each sample) and loaded onto an 8% native polyacrylamide gel. Electrophoresis was performed in 0.5x Tris-borate-EDTA (TBE) at 80–120 V on ice. The gels were stained with SYBR Gold (Thermo Fisher Sci) and visualized by Blue LED (UltraThin LED Illuminator-GelCompany).

## Mass photometry

Mass photometry experiments were conducted using a Refeyn Two$^{MP}$ mass photometer (Refeyn Ltd, Oxford, UK) to monitor the formation of the dimeric states for apo and EBM-bound Taf14$_{FL}$. Coverslips (24 mm×50 mm, Thorlabs Inc.) and silicon gaskets (Grace Bio-Labs) were cleaned with several rinses of ultrapure water and HPLC-grade isopropanol and then dried with a clean air stream. Measurements were performed at room temperature in buffer (50 mM Tris pH 7.5, 150 mM NaCl, 1 mM TCEP). 50 nM β-amylase in the same buffer was used as calibration standard (56 kDa, 112 kDa, and 224 kDa). Samples for measurements were prepared by rapid in-drop dilution. Before each measurement, 19 μl of buffer were added to a well, and the focus was set and locked. 1 μl of Taf14$_{FL}$ sample in the absence and presence of Yng1$_{EBM}$ (1:1 molar ratio) or the mixture of Yng1$_{EBM}$ and Sas3$_{EBM}$ (1:1:5 molar ratio) was added to the drop for a final concentration of 100 nM. Samples were briefly mixed, and movies were captured immediately for 60 s (2800 frames) using AcquireMP software (version 2024 R1; Refeyn Ltd). Data were processed using DiscoverMP (version 2024 R1; Refeyn Ltd).

## Circular dichroism

Circular dichroism spectra were acquired on a Jasco J-815 spectropolarimeter at 25 °C over the range of 190–250 nm in 1 nm steps with 2 s integration time, 1 nm bandwidth, 50 nm/min scan speed, and 1 mm cell length. Spectra were collected using a 1 mm cuvette. The Taf14 samples contained 12 μM Taf14$_{ET}$ without and with 12 μM Sas3$_{EBM}$, 12 μM Yng1$_{EBM}$, or the mixture of 6 μM Sas3$_{EBM}$ and 6 μM Yng1$_{EBM}$ in 25 mM sodium phosphate and 50 mM sodium fluoride buffer pH 7.5. The peptide samples contained 12 μM Sas3$_{EBM}$, 12 μM Yng1$_{EBM}$, or the mixture of 6 μM Sas3$_{EBM}$ and 6 μM Yng1$_{EBM}$ in the same buffer.

## ChIP-exo analysis

ChIP-exo peak files for Taf14, Yng1 and Sas3 generated using *Saccharomyces cerevisiae* TAP-tagged strains in BY4741 background were downloaded from the gene expression omnibus, accession number GSE147927 [https://www.ncbi.nlm.nih.gov/geo/query/acc.cgi?acc=GSE101099][29]. ChIPseeker was used to obtain genomic distributions and gene annotations of binding sites in peak list for each factor. gProfiler[38] was then used to identify GO and KEGG enrichments in the list of genes with promoters co-bound by Taf14 and Yng1.

## Site-directed mutagenesis to generate BY4741 yng1 I120D-TAP and BY4741sas3 I118D-TAP

Genomic DNA of the respective TAP-tagged wild type *Saccharomyces cerevisiae* strains under study (Supplementary Fig. 10) was used to PCR amplify full-length YNG1-TAP and SAS3-TAP from the respective strains, and these fragments were cloned into pTOPO vector for site-directed mutagenesis. For site-directed mutagenesis, the oligonucleotides pairs were designed to make the indicated amino acid codon change in the respective genes of interest. Following the QuickChange™ standard protocol as recommended by manufacturer (Agilent), the codon changes, I120D in Yng1 and I118D in Sas3, were confirmed by sequencing. The mutated yeast fragments, *yng1* I120D-TAP and *sas3* 118D-TAP, were excised from the pTOPO vector backbone by restriction enzyme digestion, and then purified. Yeast cells BY4741 yng1(I120D) and BY4741 sas3(I118D) were transformed with the purified gene fragments as previously described[39]. Following selection and genotype confirmation, the respective genomic integrants were validated by sequencing.

## Cell spotting assays

*Saccharomyces cerevisiae* strains (Supplementary Fig. 10) were cultivated in 10 ml YPD media overnight. For cell spotting assays, cells at 0.5 $OD_{600}$ were subjected to fivefold serial dilutions and spotted onto the designated plates for the indicated temperatures and days. For 6-AU studies, strains were spotted on synthetic complete (SC) media lacking uracil (Ura), or SC -Ura plates supplemented with 6-azauracil (6-AU) at 100 µg/ml. All other treatments were performed using YPD-containing plates supplemented with either 0.03% MMS, 100 mM HU, or 5 µg/ml Zeocin plates at the indicated days or temperature.

## Statistics and reproducibility

Statistical analysis was performed using GraphPad Prism 8 software. MST data are presented as average of three independent measurements ± SEM, and tryptophan fluorescence data are presented as average of three independent measurements ± SD. The apparent $K_d$ values were calculated using a 1:1 stoichiometry.

## Reporting summary

Further information on research design is available in the Nature Portfolio Reporting Summary linked to this article.

## Data availability

Coordinates and structure factors have been deposited in the Protein Data Bank under the accession code 8U77. All other relevant data supporting the key findings of this study are available within the article and its Supplementary Information. Source data are provided with this paper.

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

## Acknowledgements

This work was supported in part by NIH grants HL151334, CA252707, GM125195, GM135671 and AG067664 to T.G.K., GM118760 and CA221306 to S.D.T., GM126900 to B.D.S, and CA279317 to M.A.B. S.D.T. research is supported in part by a seed grant from the Vice Chancellor for Research and Innovation at UAMS. Mass photometry experiments were performed in the Shared Instruments Pool (RRID: SCR_018986) in the Department of Biochemistry at the University of Colorado Boulder. CU Anschutz NMR spectrometers are supported by NIH grants P30CA046934 and S10 OD025020.

## Author contributions

M.C.N., H.R., A.R., P.W., D.C.B., B.J.K. and T.M.G. performed experiments and together with A.H.E., G.Z., M.A.B., B.D.S., S.D.T. and T.G.K. analyzed the data and prepared figures. T.G.K. wrote the manuscript with input from all authors.

## Competing interests

B.D.S. is a co-founder of EpiCypher, Inc. All other authors declare no competing interests.
