## [Peer Review File · Nature Communications]

REVIEWER COMMENTS

Reviewer #1 (Remarks to the Author):

In this manuscript, Nguyen et al. define new interactions between members of the NuA3 complex, a multisubunit histone acetyltransferase complex. The authors use multiple molecular, biochemical, and structural approaches to map the interactions between Taf14 and binding partners Yng1 and Sas3. Specifically, they find that an ET domain in Taf14 recognizes a conserved binding motif known as EBM in Yng1 and Sas3, and their structural data shows that the heterodimer formation improves DNA binding of Taf14, which likely enhances its activity in vivo, as tested in mutant analysis in yeast cells. This study therefore presents new insights into the assembly of the NuA3 complex, which was not previously understood, and provides an improved mechanistic and structural understanding of the subunit interactions for this complex. The data presented are quite convincing and the conclusions are appropriately supported by the data. I only had a couple of minor concerns about the manuscript to clarify or correct a few small points in the manuscript, which are listed below. Otherwise, this work convincingly identifies key aspects of the assembly of the NuA3 complex, which regulates its function, and will be of broad interest to molecular and structural biologists interested in chromatin and gene expression regulation.

Minor concerns:

1. In Figure 4g, the authors show that Taf14 and Yng1-controlled gene regulation largely impacts genes involved in translation. Can the authors speculate on why these genes might be specifically enriched in the Taf14 and Yng1-bound set? Also, for this figure, does the size of the dots indicate the approximate number of genes? If so, a key should be added to the figure to provide an idea of the gene numbers in the relative categories. It might also be helpful to have a supplemental table of the other top ranked GO categories that are represented on this graph but not labeled.
2. Figure 1d and 1e- It would be helpful to have the schematic that indicates that domains included in the FL on panel 1d in addition to 1e (or have it on 1d only since it comes first). Otherwise, it can appear as though the data represent different constructs, even though they are both the Taf14 FL.
3. On page 7, line 160, it should state “lack” instead of “luck”.

Reviewer #2 (Remarks to the Author):

Nguyen et al. report molecular insights on the arrangement of three yeast HAT complex NuA3 subunits. Crystallography shows that the ET domain of Taf14 forms a complex with a short sequence of Yng1 and a 2:2 stoichiometry. The binding of the Yng1 peptide to the full-length Taf14 protein (and also to the ET domain) is confirmed in solution by NMR and other techniques, and the affinity is measured. They also characterize the ET domain of Taf14 as binding to a short sequence from Sas3 in solution. Cell spotting assays of wild-type and mutant yeast strains support the fact that the interactions are functionally relevant. Based on 1D NMR spectra, the authors claim that the Yng1 and Sas3 peptides form heterodimers in solution and propose a model for the spatial arrangement of the three proteins.

The overall manuscript appears to be premature for publication. Some results are relevant, but clarifications are required to evaluate them properly. Other results are preliminary and require confirmation.

Major comments:

The main finding of the work is the formation of a heterotetramer between the ET domain of Taf14 and the EBM sequence of Yng1, as seen in a crystal. MTS, fluorescence, and NMR show that the interaction occurs in solution with the full-length Taf14 and its ET domain, with low micromolar affinity. However, not all the experimental data are consistent with this interpretation. The quality of the HSQC spectra of Taf14 in the complex is much better than expected for an approximately 60 kDa complex at 25 °C using non-deuterated proteins. An increase in the linewidth of the Taf14 signals in the full-length should be noticeable compared to the smaller ET domain, and the same should be observed in the tetrameric complex as compared with the monomeric state of Taf14. But this is not what seems to occur (figures 1c, f, and supplementary figure 1). The monomeric/oligomeric state of Taf14 in solution, in the absence and presence of Yng1, should be verified by analytical ultracentrifugation or by SEC-MALS.

The proposed formation of a heterotetramer with two Taf14, one Sas3, and one Yng1 molecule (figure 6 a) relies on the observation of chemical shift changes in the 1D NMR spectrum of a mixture of the Yng1 and Sa3 peptides as compared with the spectra of the individual peptides. The relevance of this observation is difficult to evaluate because experimental details are missing. The spectra are said to be collected on peptides dissolved in 99 % D2O (line 213 and line 305), but the buffer, pH, temperature, and peptide concentration need to be specified. Lyophilized peptides dissolved in water usually yield solutions with a pH of around 3 because of their purification by

reverse phase with trifluoroacetic acid. A highly concentrated buffer, or pH adjustment, must be added to achieve a higher defined pH. Lyophilized powders contain variable amounts of trifluoroacetate and water, and the net peptide content might vary greatly from peptide to peptide stock. Because of that, the actual concentration in the solution stocks should be measured. For the chemical shift changes to be informative, the experiments should be done with peptide samples in the same buffer (for instance, by simultaneous dialysis), with a proper chemical shift reference (like DSS), and with the peptide concentration accurately measured (for instance, by absorbance at 280 nm). The highly unusual observation of a hydroxyl proton of a tyrosine sidechain, assigned to Y120 of the Sas3 molecule when bound to Yng1, could be a very strong indication of the formation of a tight complex. However, it is inconsistent with the lack of dispersion in all the NMR signals in Figures 6b and c. For instance, the aromatic ones of the same tyrosine or the methyl ones from aliphatic side chains have chemical shifts typical of random coil values in the three spectra. It would be highly extraordinary that two short linear peptides form such a specific and tight complex in an aqueous solution. For these peptide-based data to support the hypothesis of the arrangement of the three proteins, further experiments are necessary to confirm the observations. For such short peptides, it should be possible to assign their NMR spectra in H₂O, detect their conformational change upon binding to each other in an equimolar mixture, and identify the folded structure in the complex using 1H 2D TOCSY and NOESY experiments. Figure 6e, comparing AlphaFold models with the Taf14f-Yng1 crystal structure, is of very little value.

Minor comments:

Lines 98-99: If Taf14 interacting with Yng1 involves not just binding but dimerization of Taf14 and forming a heterotetramer with 2:2 stoichiometry, the reported K_d is not the dissociation constant. In the equations in lines 284 and 301, a 1:1 stoichiometry (or a single set of identical binding sites) is assumed. Perhaps the MST data were analyzed with the 2:2 stoichiometry, but this is not stated in the text. Still, the numbers obtained are useful for comparing overall affinities.

Lines 105-106: The variability in the set of individual titrations shown in Figures 1e and 1g is high, and the average “K_d” values are within the estimated uncertainties (based on the standard deviations). Therefore, the difference might not be significant.

Line 120: Figures 2 and 3 show that the structure does not resemble a head-to-toe dimer of Taf14 molecules. The upper halves of the two protomers (helix a1) contact each other, but the lower halves do not.

Lines 134-135: The interaction between amino groups of lysine residues and carboxyl groups of aspartates and glutamates are probably not hydrogen bonds but salt bridges.

Lines 176-177: What is the difference between H3K9acyl and H3K9cr peptides? What is the reason for using H3K9cr to examine the possible influence of YEATS binding H3K9acyl on Taf14 binding Yng1?

Line 179-180: I find it confusing to read “antiparallel manner” together with Figure 4e, which represents the possible arrangement of the proteins in the tetramer, with the two Taf14 chains pointing in opposite directions. The Taf14 dimer in the crystal structure (figure 2) is parallel, not antiparallel. The two Yng1 sequences form an antiparallel beta-sheet, but the two Taf14 molecules are not oriented in opposite directions.

Lines 243-245: The experimental data do not show that Taf14 sandwiches two other NuA3 complex subunits. This hypothesis is pending experimental verification.

Lines 251-263: The exact proteins used should be identified by their Uniprot codes, and each clone should be described separately, informing about non-native residues, if any, retained in the purified proteins.

Line 269: The complete sequences of the Yng1 and Sas3 peptides should be given.

Line 301: The equation needs revision: sigma should be delta, and delta should be δ_{max} .

Line 374: The PDB validation report describes 8 chains with different numbers of modeled residues. Which chains are discussed in the text and shown in Figure 2?

Line 374: The PDB validation report (page 4) contains a discrepancy that should be clarified. Residue 189 in chain C is modeled as Asn, while Leu is in yeast Taf14.

Reviewer #3 (Remarks to the Author):

The NuA3 complex, comprising the essential core subunits Sas3, Taf30, Eaf6, Nto1, and Yng1, is responsible for histone H3 acetylation. While these core subunits are vital for the assembly and function of the complex, the precise mechanism of their complex formation remains poorly understood. In this study, the authors investigated the structure of the Taf14 extra-terminal (ET) domain in complex to the ET-binding motif (EBM) in Yng1. Structural, biochemical, and mutational analyses show that dimeric EBM is sandwiched between two ET domains of Taf14. Interestingly, Sas3 also contains a similar ET-binding motif, which mediates the formation of Yng1-Sas3 heterodimer by dimerizing with Yng1 EBM. Moreover, they observed that both the interactions of Taf14 with Sas3 and Yng1 are crucial for the proper functions of the NuA3 complex in gene transcription and DNA repair. These findings offer valuable insights into the assembly of Taf14, Yng1, and Sas3 within the NuA3 complex.

Comments:

1. In 2020, Chen et al demonstrated that the Taf14 ET domain could recognize a shared motif present in various transcriptional coactivator proteins across multiple nuclear complexes, such as RSC, SWI/SNF, INO80, NuA3, TFIID, and TFIIF. They also elucidated the NMR structure of the Taf14 ET domain binding to Sth1 via a shared motif (ET-binding motif). They also analyzed the binding ability of Taf14 to the specific sequence of Sas3 containing LKVRI (Nat Commun 2020 Aug 21;11(1):4206.). What is the novel discovery in the structure of the Taf14 ET domain complex with Yng1 determined in this study?

2. What is the oligomeric state of Taf14 ET in solution? The head-to-toe dimer of Taf14 ET is mediated by the Taf14 ET or ET binding motif of Sas3 and/or Yng1? Given the hydrophobic interaction is crucial for the binding of Taf14, the binding affinity of Taf14ET to Sas3 EBM (LKVRI sequence) was moderately weaker than that of Yng1EBM, I am wondering if the Yng1EBM-Taf14 and Sas3EBM-Taf14 can form a stable heterodimer. What is the binding ability of Yng1EBM to Sas3EBM? If both Yng1EBM and Sas3EBM can form a stable heterodimer? The author should provide evidence to answer these questions.

3. The EMSA experiment was carried out with a buffer condition of 20 mM Tris-HCl pH 7.5, 25 mM NaCl, 0.2 mM ethylenediaminetetraacetic acid (EDTA), please explain why a very low salt concentration was used for DNA binding analysis. Upon examination of Figures 4a and 4b, it is not obvious that the presence of Yng1EBM enhanced the binding of Taf14 to DNA (Figures 4a and b), and the samples seem precipitated during the EMSA analysis. Moreover, the authors demonstrated that the DNA binding increase could be due to the Yng1EBM-induced dimerization of Taf14ET that creates an extended positively charged patch on the dimer surface opposite to the Yng1EBM-binding groove (Figure 4c), however, it is challenging to precisely identify the locations of the positively charged patch on the Taf14ET dimer in Figure 4c. The author can think of showing the cartoon model of the Taf14ET dimer in the same view as Figure 4c.

4. Based on the interaction pattern observed between Taf14, Yng1, and Sas3 in this study, the assembly model of the five subunits in the NuA3 complex should be revised (Figure 1a). It is proposed that Taf14 should interact with Yng1 and Sas3 simultaneously.

We thank the Editor and Reviewers for the careful review of our manuscript and the insightful and very constructive comments, which were helpful in revising and strengthening this manuscript.

Reviewer 1, Comment 1: In this manuscript, Nguyen et al. define new interactions between members of the NuA3 complex, a multisubunit histone acetyltransferase complex. The authors use multiple molecular, biochemical, and structural approaches to map the interactions between Taf14 and binding partners Yng1 and Sas3. Specifically, they find that an ET domain in Taf14 recognizes a conserved binding motif known as EBM in Yng1 and Sas3, and their structural data shows that the heterodimer formation improves DNA binding of Taf14, which likely enhances its activity in vivo, as tested in mutant analysis in yeast cells. This study therefore presents new insights into the assembly of the NuA3 complex, which was not previously understood, and provides an improved mechanistic and structural understanding of the subunit interactions for this complex. The data presented are quite convincing and the conclusions are appropriately supported by the data. I only had a couple of minor concerns about the manuscript to clarify or correct a few small points in the manuscript, which are listed below. Otherwise, this work convincingly identifies key aspects of the assembly of the NuA3 complex, which regulates its function, and will be of broad interest to molecular and structural biologists interested in chromatin and gene expression regulation.

Minor concerns:

1. In Figure 4g, the authors show that Taf14 and Yng1-controlled gene regulation largely impacts genes involved in translation. Can the authors speculate on why these genes might be specifically enriched in the Taf14 and Yng1-bound set? Also, for this figure, does the size of the dots indicate the approximate number of genes? If so, a key should be added to the figure to provide an idea of the gene numbers in the relative categories. It might also be helpful to have a supplemental table of the other top ranked GO categories that are represented on this graph but not labeled. – as suggested, we have expanded Fig. 4g legend, added the number of genes in each category, and included Suppl. Table 2 that contains all significantly enriched GO categories
2. Figure 1d and 1e- It would be helpful to have the schematic that indicates that domains included in the FL on panel 1d in addition to 1e (or have it on 1d only since it comes first). Otherwise, it can appear as though the data represent different constructs, even though they are both the Taf14 FL. – we have removed this schematic for clarity
3. On page 7, line 160, it should state “lack” instead of “luck”. – corrected

Reviewer 2, Comment 1: Nguyen et al. report molecular insights on the arrangement of three yeast HAT complex NuA3 subunits. Crystallography shows that the ET domain of Taf14 forms a complex with a short sequence of Yng1 and a 2:2 stoichiometry. The binding of the Yng1 peptide to the full-length Taf14 protein (and also to the ET domain) is confirmed in solution by NMR and other techniques, and the affinity is measured. They also characterize the ET domain of Taf14 as binding to a short sequence from Sas3 in solution. Cell spotting assays of wild-type and mutant yeast strains support the fact that the interactions are functionally relevant. Based on 1D NMR spectra, the authors claim that the Yng1 and Sas3 peptides form heterodimers in solution and propose a model for the spatial arrangement of the three proteins.

The overall manuscript appears to be premature for publication. Some results are relevant, but clarifications are required to evaluate them properly. Other results are preliminary and require confirmation.

Major comments:

The main finding of the work is the formation of a heterotetramer between the ET domain of Taf14 and the EBM sequence of Yng1, as seen in a crystal. MTS, fluorescence, and NMR show that the interaction occurs in solution with the full-length Taf14 and its ET domain, with low micromolar affinity. However, not all the experimental data are consistent with this interpretation. The quality of the HSQC spectra of Taf14 in the complex is much better than expected for an approximately 60 kDa complex at 25 °C using non-deuterated proteins. An increase in the linewidth of the Taf14 signals in the full-length should be noticeable compared to the smaller ET domain, and the same should be observed in the tetrameric complex as compared with the monomeric state of Taf14. But this is not what seems to occur (figures 1c, f, and supplementary figure 1). The monomeric/oligomeric state of Taf14f in solution, in the absence and presence of Yng1, should be verified by analytical ultracentrifugation or by SEC-MALS.

Author's response: as suggested, we have assessed the oligomeric state of Taf14FL +/- Yng1EBM using mass photometry (shown in Suppl. Fig. 3). The following sentence has been revised on page 6: "In the crystal structure, Taf14_{ET} forms a head-to-toe dimer, which creates a deep, elongated binding groove for dimeric Yng1_{EBM}, and mass photometry data indicate that the Taf14_{FL}-Yng1_{EBM} complex can also form a dimer in solution (Suppl. Fig. 3)."

We note that HSQC spectra of Taf14FL were collected on a 900 MHz spectrometer, and of the ET domain– on a 600 MHz spectrometer.

Reviewer 2, Comment 2: The proposed formation of a heterotetramer with two Taf14, one Sas3, and one Yng1 molecule (figure 6 a) relies on the observation of chemical shift changes in the 1D NMR spectrum of a mixture of the Yng1 and Sa3 peptides as compared with the spectra of the individual peptides. The relevance of this observation is difficult to evaluate because experimental details are missing. The spectra are said to be collected on peptides dissolved in 99 % D₂O (line 213 and line 305), but the buffer, pH, temperature, and peptide concentration need to be specified. Lyophilized peptides dissolved in water usually yield solutions with a pH of around 3 because of their purification by reverse phase with trifluoroacetic acid. A highly concentrated buffer, or pH adjustment, must be added to achieve a higher defined pH. Lyophilized powders contain variable amounts of trifluoroacetate and water, and the net peptide content might vary greatly from peptide to peptide stock. Because of that, the actual concentration in the solution stocks should be measured. For the chemical shift changes to be informative, the experiments should be done with peptide samples in the same buffer (for instance, by simultaneous dialysis), with a proper chemical shift reference (like DSS), and with the peptide concentration accurately measured (for instance, by absorbance at 280 nm). The highly unusual observation of a hydroxyl proton of a tyrosine sidechain, assigned to Y120 of the Sas3 molecule when bound to Yng1, could be a very strong indication of the formation of a tight complex. However, it is inconsistent with the lack of dispersion in all the NMR signals in Figures 6b and c. For instance, the aromatic ones of the same tyrosine or the methyl ones from aliphatic side chains have chemical shifts typical of random coil values in the three spectra. It would be highly extraordinary that two short linear peptides form such a specific and tight complex in an aqueous solution. For these peptide-based data to support the hypothesis of the arrangement of the three proteins, further experiments are necessary to confirm the observations. For such short peptides, it should be possible to assign their NMR spectra in H₂O, detect their conformational change upon binding to each other in an equimolar mixture, and identify the folded structure in the complex using 1H 2D TOCSY and NOESY experiments. Figure 6e, comparing AlphaFold models with the Taf14f-Yng1 crystal structure, is of very little value.

Author's response: we very much appreciate this point. We carried out numerous 1D NMR experiments using different conditions, varying ionic strength and pH in both D₂O and H₂O. Indeed, NMR spectra of these peptides are very sensitive to both ionic strength/salt concentration and pH, which is likely because of the presence of 11 ionizable residues in these

peptides. In addition, searching literature regarding contribution of hydrogen bonds and salt bridges, we see that these peptides are too short to unambiguously measure changes in enthalpy. To avoid overstatement, we have removed panels (c) and (d) from Fig. 6 (keeping AF models as they are of help here), and instead used CD, EMSA and mass photometry to validate the complexes. We found that the mixture of Yng1_{EBM} and Sas3_{EBM} stabilizes the structure of the complex to the same extent as individual Yng1_{EBM} and Sas3_{EBM} stabilize it. We also show that the DNA binding activity of Taf14_{ET} is augmented by either the mixture of EBM or Yng1_{EBM}, and comparable molecular mass of Taf14_{ET} was measured in solution in the presence of Yng1_{EBM} or the mixture of EBM (Suppl. Figs. 3, 8 and 9).

Reviewer 2, Comment 3: Minor comments:

Lines 98-99: If Taf14 interacting with Yng1 involves not just binding but dimerization of Taf14 and forming a heterotetramer with 2.2 stoichiometry, the reported K_d is not the dissociation constant. In the equations in lines 284 and 301, a 1:1 stoichiometry (or a single set of identical binding sites) is assumed. Perhaps the MST data were analyzed with the 2:2 stoichiometry, but this is not stated in the text. Still, the numbers obtained are useful for comparing overall affinities. – all binding data were analyzed using a 1:1 stoichiometry for comparison. We have added to Figs. 1, 3, 4, and 5 legends and in methods that the apparent K_d values were calculated using a 1:1 stoichiometry.

Lines 105-106: The variability in the set of individual titrations shown in Figures 1e and 1g is high, and the average “K_d” values are within the estimated uncertainties (based on the standard deviations). Therefore, the difference might not be significant. – although the values are not markedly different, they do not overlap, and we have observed a similar reduction in binding of other ligands of ET, Ino80 and Snf5, in context of the full-length protein. Due to binding of the linker region of Taf14 to ET, the linker is not freely available for binding to DNA, and reciprocally, binding of Ino80 and Snf5 to the ET domain is also reduced. We describe this in detail in Klein, Nat Com, 2022 and Nguyen, BBA, 2023.

Line 120: Figures 2 and 3 show that the structure does not resemble a head-to-toe dimer of Taf14 molecules. The upper halves of the two protomers (helix a1) contact each other, but the lower halves do not. – the two molecules of Taf14 ET are related by two-fold rotation symmetry (180°) and are arranged in a head-to-toe position. We have added Suppl. Fig. 4 to make it clearer.

Lines 134-135: The interaction between amino groups of lysine residues and carboxyl groups of aspartates and glutamates are probably not hydrogen bonds but salt bridges. – this sentence has been revised to: “The side chain amino groups of K115, K119 and K124 of Yng1_{EBM} form salt bridges with the carboxyl groups...”

Lines 176-177: What is the difference between H3K9acyl and H3K9cr peptides? What is the reason for using H3K9cr to examine the possible influence of YEATS binding H3K9acyl on Taf14 binding Yng1? – H3K9cr and H3K9ac (altogether are H3K9acyl) are physiological ligands of the YEATS domain. We have changed acyl to cr in the text for clarity.

Line 179-180: I find it confusing to read “antiparallel manner” together with Figure 4e, which represents the possible arrangement of the proteins in the tetramer, with the two Taf14 chains pointing in opposite directions. The Taf14 dimer in the crystal structure (figure 2) is parallel, not antiparallel. The two Yng1 sequences form an antiparallel beta-sheet, but the two Taf14 molecules are not oriented in opposite directions. – the Taf14 dimer in the crystal structure is in an antiparallel orientation, with the N- and C-termini of the two protomers being oriented in the opposite direction (please see Suppl. Fig. 4).

Lines 243-245: The experimental data do not show that Taf14 sandwiches two other NuA3 complex

subunits. This hypothesis is pending experimental verification. – this sentence has been revised to: “Our structural and functional analyses suggest a model in which the Taf14 subunit may bridge Yng1 and Sas3 through the formation of a heterodimeric assembly.”

Lines 251-263: The exact proteins used should be identified by their Uniprot codes, and each clone should be described separately, informing about non-native residues, if any, retained in the purified proteins. – Uniprot codes and information about extra vector-derived residues have been added in methods, page 11.

Line 269: The complete sequences of the Yng1 and Sas3 peptides should be given. – complete sequences of the peptides are now added in methods, page 12.

Line 301: The equation needs revision: sigma should be delta, and delta should be deltamax. – this glitch has been fixed, thank you

Line 374: The PDB validation report describes 8 chains with different numbers of modeled residues. Which chains are discussed in the text and shown in Figure 2? – chains A, B, C, and D were used

Line 374: The PDB validation report (page 4) contains a discrepancy that should be clarified. Residue 189 in chain C is modeled as Asn, while Leu is in yeast Taf14. – this residue has been corrected in the revised validation report

Reviewer 3, Comment 1: The NuA3 complex, comprising the essential core subunits Sas3, Taf30, Eaf6, Nto1, and Yng1, is responsible for histone H3 acetylation. While these core subunits are vital for the assembly and function of the complex, the precise mechanism of their complex formation remains poorly understood. In this study, the authors investigated the structure of the Taf14 extra-terminal (ET) domain in complex to the ET-binding motif (EBM) in Yng1. Structural, biochemical, and mutational analyses show that dimeric EBM is sandwiched between two ET domains of Taf14. Interestingly, Sas3 also contains a similar ET-binding motif, which mediates the formation of Yng1-Sas3 heterodimer by dimerizing with Yng1 EBM. Moreover, they observed that both the interactions of Taf14 with Sas3 and Yng1 are crucial for the proper functions of the NuA3 complex in gene transcription and DNA repair. These findings offer valuable insights into the assembly of Taf14, Yng1, and Sas3 within the NuA3 complex.

Comments:

1. In 2020, Chen et al demonstrated that the Taf14 ET domain could recognize a shared motif present in various transcriptional coactivator proteins across multiple nuclear complexes, such as RSC, SWI/SNF, INO80, NuA3, TFIID, and TFIIF. They also elucidated the NMR structure of the Taf14 ET domain binding to Sth1 via a shared motif (ET-binding motif). They also analyzed the binding ability of Taf14 to the specific sequence of Sas3 containing LKVR1 (Nat Commun 2020 Aug 21;11(1):4206.). What is the novel discovery in the structure of the Taf14 ET domain complex with Yng1 determined in this study?

Author’s response: both Chen et al Nat Com, 2020 studies and our previous work, Klein et al Nat Com, 2022 report the structures of a monomeric ET domain in complex with Sth1 or Taf2, which help to understand functions of the ATP-dependent chromatin-remodeling complex RSC and the general transcription factors complex TFIID. We cite both papers in the introduction and in the results section, they are references 24 and 25, and added ref 24 to the paragraph describing Sas3.

The findings that, within the histone acetyltransferase complex NuA3, the Taf14 subunit interacts with EBM from two other complex subunits, Yng1 and Sas3, and that dimeric EBM is bound by two ET domains suggest a mechanism which is distinctly different from the mechanism observed for the association of Taf14 with Taf2, in which Taf2 is sandwiched between the ET and YEATS domains of Taf14.

Reviewer 3, Comment 2: What is the oligomeric state of Taf14 ET in solution? The head-to-toe dimer of Taf14 ET is mediated by the Taf14 ET or ET binding motif of Sas3 and/or Yng1? Given the hydrophobic interaction is crucial for the binding of Taf14, the binding affinity of Taf14_{ET} to Sas3 EBM (LKVRV sequence) was moderately weaker than that of Yng1EBM, I am wondering if the Yng1EBM-Taf14 and Sas3EBM-Taf14 can form a stable heterodimer. What is the binding ability of Yng1EBM to Sas3EBM? If both Yng1EBM and Sas3EBM can form a stable heterodimer? The author should provide evidence to answer these questions.

Author's response: we have assessed the oligomeric state of Taf14_{FL} +/-Yng1 using mass photometry (shown in Suppl. Fig. 3) (an isolated ET domain is too small to be analyzed). The following sentence has been revised on page 6: "In the crystal structure, Taf14_{ET} forms a head-to-toe dimer, which creates a deep, elongated binding groove for dimeric Yng1_{EBM}, and mass photometry data indicate that the Taf14_{FL}-Yng1_{EBM} complex can also form a dimer in solution (Suppl. Fig. 3)."

Our attempts to measure a K_d for the association between FAM-labeled Yng1 and unlabeled Sas3 peptides by MST were unsuccessful- the peptides are too short.

Reviewer 3, Comment 3: The EMSA experiment was carried out with a buffer condition of 20 mM Tris-HCl pH 7.5, 25 mM NaCl, 0.2 mM ethylenediaminetetraacetic acid (EDTA), please explain why a very low salt concentration was used for DNA binding analysis. Upon examination of Figures 4a and 4b, it is not obvious that the presence of Yng1EBM enhanced the binding of Taf14 to DNA (Figures 4a and b), and the samples seem precipitated during the EMSA analysis. Moreover, the authors demonstrated that the DNA binding increase could be due to the Yng1EBM-induced dimerization of Taf14_{ET} that creates an extended positively charged patch on the dimer surface opposite to the Yng1EBM-binding groove (Figure 4c), however, it is challenging to precisely identify the locations of the positively charged patch on the Taf14_{ET} dimer in Figure 4c. The author can think of showing the cartoon model of the Taf14_{ET} dimer in the same view as Figure 4c.

Author's response: we have repeated EMSA experiments (shown in Suppl. Fig. 6) and confirmed that in the presence of the Yng1 peptide, binding of the ET domain to DNA is enhanced. Because 601 DNA contains ~14 major/minor grooves, it is often difficult to see strong ladder bands for the multiple 601-protein complexes, still the bands are present at the DNA:Taf14 ratio of 1:200 and 1:500 (Fig. 4a). The bands are more visible at low salt concentrations, which is needed when the relative (+/- Yng1) DNA binding activities are compared. Higher 601 DNA:protein ratio leads to the formation of large complexes that remain in well (also observed for many other protein-601 DNA complexes, such as MORF (Nat Com, 2023), p300 (Nat Com, 2021), MORC3/4 (Nat Com, 2020), JADE (NSMB, 2024)).

Please note that the shadow (not precipitation) is caused by reflection/refraction of light through the gel boundary at the bottom of each loading well. This shadow is observed in all wells, even in wells without proteins.

Fig. 4c has been revised to better illustrate the position of the positively charged patch with respect to the Yng1-binding site.

Reviewer 3, Comment 4: Based on the interaction pattern observed between Taf14, Yng1, and Sas3 in this study, the assembly model of the five subunits in the NuA3 complex should be revised (Figure 1a). It is proposed that Taf14 should interact with Yng1 and Sas3 simultaneously. – this figure has been modified as suggested, thank you

REVIEWERS' COMMENTS

Reviewer #1 (Remarks to the Author):

The comments I provided to the authors were addressed in the revised manuscript. I have no additional concerns regarding the work presented in this manuscript.

Reviewer #2 (Remarks to the Author):

Nguyen et al. present in the revised article additional experimental data, figures, and clarifications that make it a strong one suitable for publication. I suggest the following minor changes that will, in my opinion, improve it.

Indicate in the main text that the Taf14FL protein by itself forms dimers in solution.

Indicate in the methods which kind of “column chromatography” was used for further purification of the proteins, how protein and peptide concentration were measured, and which proteins were studied at 900 or 600 MHz.

Amend the text inside Suppl. Figure 3: it says “Taf14ET” but the caption and the main text say “Taf14FL”.

Indicate in the caption of Suppl. Figure 3 the expected molecular masses for the Taf14FL protein and its complex with Yng1EBM.

Reviewer #3 (Remarks to the Author):

The authors have made a certain response to the request for revising. I have no additional questions or concerns for the authors.

#Response to comments of Reviewer 2

Indicate in the main text that the Taf14FL protein by itself forms dimers in solution. – included in methods on page 15 and in the expanded Suppl. Fig. 3 legend

Indicate in the methods which kind of “column chromatography” was used for further purification Of the proteins, how protein and peptide concentration were measured, and which proteins were studied at 900 or 600 MHz. – added on pages 11, 12 and 13

Amend the text inside Suppl. Figure 3: it says “Taf14ET” but the caption and the main text say “Taf14FL”. – corrected

Indicate in the caption of Suppl. Figure 3 the expected molecular masses for the Taf14FL protein and its complex with Yng1EBM. – added